# Lysophosphatidic Acid Receptor 1- and 3-Mediated Hyperalgesia and Hypoalgesia in Diabetic Neuropathic Pain Models in Mice

**DOI:** 10.3390/cells9081906

**Published:** 2020-08-16

**Authors:** Hiroshi Ueda, Hiroyuki Neyama, Yosuke Matsushita

**Affiliations:** 1Department of Molecular Pharmacology, Kyoto University Graduate School of Pharmaceutical Sciences, Yoshida Shimoadachi-cho, Sakyo-ku, Kyoto 606-8501, Japan; 2RIKEN Center for Biosystems Dynamics Research, Kobe 650-0047, Japan; hiroyuki.neyama@riken.jp; 3Division of Genome Medicine, Institute of Advanced Medical Sciences, Tokushima University, Tokushim 770-8501, Japan; y-matsushita@genome.tokushima-u.ac.jp

**Keywords:** hypoalgesia, lysophosphatidic acid receptor, streptozotocin, high-fat diet, leptin receptor, pathogenic

## Abstract

Lysophosphatidic acid (LPA) signaling is known to play key roles in the initiation and maintenance of various chronic pain models. Here we examined whether LPA signaling is also involved in diabetes-induced abnormal pain behaviors. The high-fat diet (HFD) showing elevation of blood glucose levels and body weight caused thermal, mechanical hyperalgesia, hypersensitivity to 2000 or 250 Hz electrical-stimulation and hyposensitivity to 5 Hz stimulation to the paw in wild-type (WT) mice. These HFD-induced abnormal pain behaviors and body weight increase, but not elevated glucose levels were abolished in LPA_1_^−/−^ and LPA_3_^−/−^ mice. Repeated daily intrathecal (i.t.) treatments with LPA_1/3_ antagonist AM966 reversed these abnormal pain behaviors. Similar abnormal pain behaviors and their blockade by daily AM966 (i.t.) or twice daily Ki16425, another LPA_1/3_ antagonist was also observed in db/db mice which show high glucose levels and body weight. Furthermore, streptozotocin-induced similar abnormal pain behaviors, but not elevated glucose levels or body weight loss were abolished in LPA_1_^−/−^ and LPA_3_^−/−^ mice. These results suggest that LPA_1_ and LPA_3_ play key roles in the development of both type I and type II diabetic neuropathic pain.

## 1. Introduction

Neuropathic pain (NeuP) occurs when neurons in the pain pathway from primary afferent nerves to the central nervous system are damaged [1]. NeuP also occurs as a secondary symptom in diabetes, cancer, and herpes zoster infection, or as a side effect when patients are treated by chemotherapeutics [2,3,4]. It is often characterized by abnormal sensory perception of pain such as allodynia and hyperalgesia, which are represented as pain perception on exposure to tactile stimuli and exaggerated pain sensations to mildly noxious stimuli, respectively [4,5]. Due to the delayed onset and persistence after the healing of the original causes of NeuP, proper diagnosis and early treatment are difficult. Diabetic peripheral neuropathy (DPN) is one of the most frequent (approximately 50%) complications of type I and type II diabetes mellitus. Almost 10%–30% of diabetic patients develop neuropathic pain (DPNP), which shows abnormal pain sensations including hyperalgesia, allodynia, and hypoalgesia/hypoesthesia/numbness [6,7,8,9,10]. However, it remains elusive how the hypoalgesia is paradoxically accompanied with tonic and ongoing pain in diabetic patients. Furthermore, pregabalin and duloxetine are now used worldwide for the management of DPNP, but 10%–20% of patients discontinue their long-term use due to adverse events such as dizziness, peripheral edema, weight gain, somnolence, constipation, and hypotension [11]. Thus, the novel therapeutic strategies to treat both hyperalgesia and hypoalgesia in DPNP have been demanded.

We discovered that lysophosphatidic acid (LPA) receptor 1 (LPA_1_) plays a key role in the initiation of peripheral nerve injury-induced NeuP through a demyelination of dorsal root and Ca_v_α2δ1 upregulation in dorsal root ganglion, which may cause a cross-talk among noxious and innocuous fibers (a mechanism underlying allodynia) and facilitate spinal cord pain transmission (hyperalgesia) [12]. Subsequent studies have revealed that LPA_1_ and LPA_3_ signaling also contribute to the maintenance of NeuP through microglia-mediated amplification of LPA production in the spinal cord and subsequent astrocyte-mediated chemokine production [13,14,15]. The involvement of LPA_1_ and LPA_3_ signaling has been also evidenced in various chronic pain models, such as paclitaxel-induced NeuP, central post-stroke-induced central NeuP and some fibromyalgia-like mouse models [16,17,18,19], suggesting that LPA receptor signaling would be common in various chronic pain mechanisms and compounds to suppress the LPA receptor signaling would be promising for the therapy of NeuP [20,21]. In the present study, we report the involvement of LPA_1_ and LPA_3_ signaling in both positive and negative signs of DPNP using experimental type I and type II DPN models and discuss their possibility as therapeutic targets.

## 2. Materials and Methods

### 2.1. Drugs

Drugs were administered through subcutaneous (s.c.; 100 μL/10 g), intrathecal (i.t.; 5 μL), and intravenous (i.v.; 100 μL/10 g) routes. The i.t. injection was administered into the space between the spinal L5 and L6 segments, according to the previously described method [22]. AM966(2-[4-[4-[4-[[(1R)-1-(2-chlorophenyl)ethoxy]carbonylamino]-3-methyl-1,2-oxazol-5-yl]phenyl]phenyl]acetic acid) purchased from Cayman Chemical (Ann Arbor, MI, USA), is an LPA receptor 1- and 3 (LPA_1_ and LPA_3_)-antagonist [23], and was first dissolved in artificial cerebrospinal fluid (a-CSF; 125 mM NaCl, 3.8 mM, KCl, 1.2 mM KH_2_PO_4_, 26 mM NaHCO_3_, 10 mM glucose, pH 7.4) with 0.1% dimethyl sulfoxide (DMSO) for intrathecal injection. Ki-16425, an LPA_1_ and LPA_3_ antagonist [24] was generously provided by Kirin Brewery Co. (Takasaki, Japan), and was dissolved in sesame oil (Sigma, St. Louis, MO, USA) immediately before administration. Ki-16425 was dissolved in physiological saline and used for subcutaneous injection. Fresh streptozotocin (STZ; Wako, Osaka, Japan) was dissolved in saline, adjusted to pH 4.5 in 0.1 N citrate buffer, and used for i.v. injection. The doses of AM966 and Ki-16425 were selected from several preliminary trials and the previous experiences in different sets of chronic pain studies, in which they produced an efficient blockade by repeated treatments, but did not cause toxic actions [15,18].

### 2.2. Animals

All experiments were approved by the Nagasaki University Animal Care Committee (Approval Number: 1707311398-4) and complied with the recommendations of the International Association for the Study of Pain [25]. All studies using animals are reported in accordance with the ARRIVE guidelines [26,27,28] and we investigated the drug effects using a blinded testing method. In the studies using high-fat diet (HFD)- and STZ-induced diabetic mice, six-week-old male C57BL/6J (wild-type/WT) mice purchased from TEXAM (Nagasaki, Japan), and LPA_1_ and LPA_3_-knockout (LPA_1_^−/−^ and LPA_3_^−/−^, respectively) mice (gift from Prof Jerold Chun, Sanford Burnham Prebys Medical Discovery Institute, La Jolla, CA, USA) were used. In a study using spontaneous type II diabetic mice, leptin receptor-deficient homozygotes, eight-weeks-old BKS.Cg-+Lepr^db^/+Lepr^db^/Jcl (db/db) mice, its heterozygotes, BKS.Cg-m+/+Lepr^db^/Jcl mice, and WT BKS.Cg-m+/m+/Jcl, were purchased from CLEA Japan, Inc. (Tokyo, Japan). Animals were maintained at 22 ± 3 °C and 55 ± 5% relative humidity, with a 12 h light/dark cycle (light on 8:00 A.M. to 8:00 P.M.) and free access to a standard laboratory normal diet (ND) (CE-2: 339.9 kcal/100 g: 25.1% protein and 4.51% fat, CLEA Japan, Inc., Tokyo, Japan) or high-fat diet 32 (507.6 kcal/100 g: 25.5 g/100 g protein and 32 g/100 g fat, CLEA Japan), and tap water. The body weights of mice tested varied within 10% in each experimental Section 2.2.

More specifically, the HFD-induced type II diabetic model was established according to the previously reported protocol [29], in which mice were fed either the ND or HFD starting at 10 weeks of age. To avoid infection, mice were maintained in a clean cage throughout the experiments. The HFD chow was replaced every other day by fresh chow, which was thawed to the room temperature after removal from a −20 °C freezer. On the other hand, the STZ-induced diabetic model was established as previously reported [30,31,32], where a single i.v. injection of 200 mg/kg STZ was used. The schematized study design of diabetic mouse models is shown in Figure 1. Details of serum metabolic parameters, including cholesterol, triglyceride, and free fatty acids in mice treated with the same recipe of HFD and the same source of leptin-deficient mice have been previously reported by other investigators [33,34]. The information of plasma insulin levels and sorbitol contents and Na^+^-K^+^-ATPase activity in peripheral nerve cells of diabetic mice using the same protocol of STZ-treatment have been also previously reported [30].

### 2.3. Thermal Paw Withdrawal Test

The thermal pain threshold (Hargreaves test) was evaluated based on the latency of paw withdrawal upon application of a thermal stimulus [18,35]. Unanesthetized animals were placed in plexiglass cages on top of a glass sheet, and an adaptation period of 1–3 h was allowed. The thermal stimulator (IITC Inc., Woodland Hills, CA, USA) was positioned under the glass sheet, and the focus of the projection bulb was aimed exactly at the middle of the plantar surface of the animal. A mirror attached to the stimulator allowed visualization of the plantar surface. A cut-off time of 20 s was set to prevent tissue damage. Mice were placed in a plexiglass chamber on a 6 × 6 mm wire mesh grid floor and allowed to acclimatize for 1 h, as previously described [36].

### 2.4. Electrical Stimulation-Induced Paw Withdrawal (EPW) Test

EPW was performed as described previously [37,38]. Briefly, electrodes of the Neurometer^Ⓡ^ Current Perception Threshold/C (CPT/C, Neurotron Inc., Baltimore, MD, USA) were fastened to the planter and the insteps of the hind paw. Transcutaneous nerve stimuli with each of the three sine-wave pulses (5, 250, and 2000 Hz) were applied. The minimum intensity (μA) at which the mouse withdrew the paw was defined as the current threshold. Sensory fiber specificity has been characterized by electrophysiological studies using preparations of acutely isolated rat dorsal root ganglion and in vivo patch-clamp recording measuring excitatory synaptic responses evoked by transcutaneous stimuli in substantia gelatinosa neurons of the spinal dorsal horn, as reported elsewhere [39].

### 2.5. Mechanical Paw Pressure Test and von Frey Filament Test

Mice were placed on a 6 × 6 mm wire mesh grid floor and allowed to acclimatize for 1 h, as previously described [18]. Mechanical pressure was then delivered to the middle of the plantar surface of the right hind paw using an electronic digital von Frey Anesthesiometer^®^ with a rigid tip (Model 2390, 90 g probe 0.8 mm outer diameter; IITC Life Sci Inc. Woodland, CA, USA). The von Frey filament test was performed using a touch test sensory evaluator kit (BrainScience/Idea Co. Ltd., Osaka, Japan).

### 2.6. Blood Glucose Measurement

Blood glucose levels were assessed using blood collected from the tail vein and measured using the FreeStyle monitoring system (Abbott, Alameda, CA, USA).

### 2.7. Statistics

All data are expressed as the mean ± standard error of the mean (S.E.M.). Data were analyzed with GraphPad Prism 7.0 (GraphPad Software, San Diego, CA, USA) using the unpaired *t*-test, one-way analysis of variance followed by Tukey’s multiple comparisons test or Dunnett’s multiple comparisons test, two-way ANOVA with Tukey’s multiple comparisons test, or Bonferroni’s multiple comparisons test. Differences with *p*-values less than 0.05 were considered statistically significant.

## 3. Results

### 3.1. Involvement of LPA_1_ and LPA_3_ in High-Fat Diet (HFD)-Induced Abnormal Pain Behaviors

In C57BL/6J (WT) mice fed with the HFD, the increase in blood glucose levels started at 2 weeks, and significant changes were observed at 6 and 10 weeks, while no significant change was observed with ND feeding, as shown in Figure 2A.

Increased blood glucose levels were also observed in both LPA_1_^−/−^ and LPA_3_^−/−^ mice, although some differences were observed in the time course from that in WT mice. In LPA_1_^−/−^ mice, the increase was significant between 2 and 4 weeks, but declined between 6 and 8 weeks, and significant again at 10 weeks. In LPA_3_^−/−^ mice, a significant increase was observed at 2 weeks after the start of HFD feeding and remained high throughout the 10 weeks.

The paw withdrawal latency (s) in the thermal nociceptive test was significantly decreased in WT mice fed the HFD, as shown in Figure 2B. A significant decrease was observed at 4 weeks and later after the start of HFD feeding, and this thermal hyperalgesia was abolished in LPA_1_^−/−^ and LPA_3_^−/−^ mice. On the other hand, the threshold to demonstrate nociceptive paw withdrawal behaviors following electrical stimulation was investigated via the EPW test using the Neurometer^Ⓡ^, in which nociceptive paw withdrawal responses to electrical stimulation of the paw at different frequencies were evaluated, as shown in Figure 3. Electrical stimulation at 2000 Hz is reported to stimulate Aβ-fibers, thereby causing an innocuous, but unpleasant, and vibratory sensation to the experimenter’s finger and eliciting paw withdrawal behaviors in the model [37]. The average threshold currents of 2000 Hz inducing withdrawal behavior in WT mice receiving ND and HFD were both approximately 320 μA, as shown in Figure 3A. The threshold significantly decreased following HFD at 4 weeks and later. HFD-induced hypersensitivity at 4–8 weeks was completely abolished in LPA_1_^−/−^ and LPA_3_^−/−^ mice. The reason why there was a lack of blockade in LPA_1_^−/−^ and LPA_3_^−/−^ mice at 10 weeks remains elusive, but it may be related to the sensitization of Aβ-fiber-related responses by repeated trials of experiments. Comparable results were observed following 250 Hz in terms of the time course of hyperalgesia and its blockade in LPA_1_^−/−^ and LPA_3_^−/−^ mice (Figure 3B).

In contrast, HFD increased the threshold of 5 Hz electrical stimulation (Figure 3C). Significant hypoalgesia was observed at 6 weeks and later and was abolished in LPA_1_^−/−^ and LPA_3_^−/−^ mice. However, there were no changes in blood glucose levels, nociceptive threshold in the thermal test and EPW tests in normal diet (ND)-fed LPA_1_^−/−^ and LPA_3_^−/−^ mice for 10 weeks (Figure 2 and Figure 3).

On the other hand, as shown in (Appendix A), HFD-treatment caused significant mechanical allodynia in the von Frey filament test at 12 weeks after the start of ND or HFD-treatments, while this allodynia was abolished in LPA_1_^−/−^ mice, and partially reversed in LPA_3_^−/−^ mice. Similarly, the mechanical hyperalgesia using digital von Frey Anesthesiometer^Ⓡ^ test was also reversed both in LPA_1_^−/−^ and LPA_3_^−/−^ mice (Appendix A**)**. However, there were no significant differences between ND-fed WT and LPA_1_^−/−^ or LPA_3_^−/−^ mice in both von Frey filament and digital von Frey tests.

### 3.2. Complete Reversal of HFD-Induced Abnormal Pain Behaviors by an LPA_1/3_ Antagonist

To investigate the roles of the spinal cord LPA_1_ and LPA_3_ in the maintenance of diabetic neuropathic abnormal pain behaviors, we intrathecally (i.t.) administered the LPA_1_ and LPA_3_ antagonist AM966 to HFD-treated mice. The repeated once-daily i.t. treatments with AM966 for 2 weeks, from 12 to 14 weeks after the start of ND or HFD, demonstrated no significant change in body weight (Figure 4A). The repeated treatments gradually reversed the thermal hyperalgesia, and significant and complete recovery was observed as early as day 7 and at days 10 and 14, respectively, (Figure 4B). Similar and complete recoveries in the hypersensitivity and hyperalgesia to the electrical stimulation at 2000 Hz and 250 Hz, respectively, were observed by repeated treatments with AM966 (Figure 4C,D). Additionally, hypoalgesia to the electrical stimulation at 5 Hz was also completely reversed by AM966 treatment (Figure 4E).

### 3.3. Involvement of LPA_1_ and LPA_3_ in HFD-Induced Type II Diabetic Obesity

The body weight of WT mice fed with HFD showed a gradual increase as the treatment period increased through 10 weeks and a significant increase was observed at the start of 3 weeks, while no increase was observed in mice fed the ND (Appendix A). HFD-induced increase in body weight was largely suppressed both in LPA_1_^−/−^ and LPA_3_^−/−^ mice (Appendix A). Only one mouse each from LPA_1_^−/−^ and LPA_3_^−/−^ mouse group showed a uniquely high level of body weight, but they showed regular nociceptive thresholds and blood glucose levels (Figure 2A). To see more details of obesity in HFD-fed mice, we used another set of 5 mice at 12 weeks after the start of ND/HFD. As shown in Appendix A, WT mice fed with HFD for 12 weeks demonstrated marked obesity, with heavy abdominal adipose tissue, when compared with ND mice. However, apparent obesity was not observed in LPA_1_^−/−^ mice fed with HFD (Appendix A), although some obesity was observed in LPA_3_^−/−^ mice fed the HFD (Appendix A). As shown in (Appendix A), the epididymal adipose tissue weight was markedly increased in HFD-fed WT mice, and the increase was significantly inhibited in LPA_1_^−/−^, but not in LPA_3_^−/−^ mice. Similar results were also observed with perirenal adipose weights (Appendix A). Furthermore, the liver weight was increased in HFD-fed WT mice, but the increase was completely or partially inhibited in LPA_1_^−/−^ or LPA_3_^−/−^ mice, respectively, (Appendix A). On measuring food, a weak, but insignificant decrease was observed in WT, LPA_1_^−/−^, and LPA_3_^−/−^ mice fed the HFD (Appendix A). The water intake also decreased by HFD, and the change was significant at all periods between 0 and 10 weeks, except for the period 4–6 weeks in WT mice, as shown in (Appendix A). Decreased water intake was also observed in LPA_1_^−/−^ and LPA_3_^−/−^ mice, but the differences were not significant.

### 3.4. Complete Reversal of Abnormal Pain Behaviors in db/db Mice by LPA_1/3_ Antagonists

Blood glucose levels of WT (+/+), leptin receptor heterozygotic (+/−) and homozygotic (db/db) mice treated with vehicle were 107.5 ± 5.6 (*n* = 6), 116.8 ± 3.9 (*n* = 6) and 358.5 ± 17.3 mg/dL (*n* = 6), respectively, while body weight of these mice was 21.9 ± 0.4 (*n* = 6), 27.5 ± 0.5 (*n* = 6) and 42.5 ± 0.23 g (*n* = 6), respectively, as shown in Figure 5A. There was a marginal or marked increase of body weight in +/− or db/db mice, compared with +/+ mice, while there was no significant difference in glucose levels between +/− and +/+, though a marked increase was observed in db/db mice.

A study evaluating abnormal pain behaviors in db/db, +/− and WT mice, and their blockade by LPA_1_ and LPA_3_ antagonists, was performed. As shown in Figure 5B, the db/db mice showed thermal hyperalgesia to the levels observed in HFD-treated mice, while no significant hyperalgesia observed in the +/− or WT mice. In db/db mice, repeated Ki16425 treatments significantly reversed the basal threshold, which was evaluated in the morning before the Ki16425 administration, while the treatments demonstrated no effect in +/− or WT mice. As in the HFD-models, db/db mice showed significant hypersensitivity, hyperalgesia, and hypoalgesia in the experiments with 2000, 250, and 5 Hz electrical stimulation to the paw, respectively, and repeated treatments with Ki16425 significantly reversed all these abnormal pain behaviors (Figure 5C–E).

Comparable observations were observed when AM966 (i.t.) was treated once daily for 2 weeks, as shown in Figure 6A–E. AM966 treatments had no effect on the body weight of WT and db/db mice, but showed complete reversal in thermal hyperalgesia, and hypersensitivity at 2000 Hz, hyperalgesia at 250 Hz and hypoalgesia at 5 Hz in the EPW test.

### 3.5. LPA_1_ and LPA_3_-Mediated Thermal Hyperalgesia in a Streptozotocin (STZ)-Induced Diabetic Mouse Model

The type I diabetic mouse model was established following the administration of 200 mg/kg (i.v.) of ‘freshly’ prepared STZ solution, which is known to inhibit insulin secretion by disturbing the transcription and replication of DNA in pancreatic β cells highly expressing glucose transporter II [40,41], as previously reported [31,32]. The control mice demonstrated a slow gain in body weight from day 10 to day 49, while body weights of STZ-mice slowly decreased from day 7 to day 49 (Appendix A). Similar STZ-induced changes in body weight were also observed in LPA_1_^−/−^ and LPA_3_^−/−^ mice (Appendix A). As shown in (Appendix A), a significant increase was observed in blood glucose levels from approximately 150 to 450 mg/dL on day 3 following the STZ administration, and it lasted through day 49. Similar STZ-induced changes in blood glucose levels were also observed in LPA_1_^−/−^ and LPA_3_^−/−^ mice (Appendix A**)**, but no significant difference from the WT levels.

When the thermal nociceptive threshold was measured, significant hyperalgesia was observed at day 3, reached a maximum at day 21, and remained significantly low through day 49 in WT mice, as shown in Figure 6. However, in LPA_1_^−/−^ mice, there was no significant change in the thermal hyperalgesia from day 3 to day 21, compared with WT threshold, but was largely reversed from day 28. Similar results were also observed in LPA_3_^−/−^ mice (Figure 7).

### 3.6. Sensory Fiber-Specific Changes in the Nociceptive Threshold in the STZ-Model

The threshold to induce paw withdrawal behavior following 2000 Hz stimulation to the right paw was approximately 300 μA in WT mice through day 49. After STZ was administered, the threshold significantly decreased on day 3, reached a maximum on day 10, and was maintained at decreased levels through day 49, as shown in (Figure 8A). In LPA_1_^−/−^ and LPA_3_^−/−^ mice, a significant decrease in the threshold was observed from day 3 to 21 but was restored to the normal levels from day 28 through day 49. Similar changes were also observed following electrical stimulation at 250 Hz (Figure 8B). However, a large difference was observed with electrical stimulation at 5 Hz, as shown in (Figure 8C). A significant increase in the nociceptive paw withdrawal threshold was observed from day 21 through day 49 after the STZ treatment, while no change was observed in control mice. The unique hypoalgesia/hypoesthesia or numbness observed at the late stage was completely abolished in LPA_1_^−/−^ and LPA_3_^−/−^ mice (Figure 8C). There was no change in each threshold at 2000, 250, and 5 Hz in LPA_1_^−/−^ and LPA_3_^−/−^ mice throughout 49 days after the vehicle was treated instead of STZ (Figure 8A–C).

## 4. Discussion

Diabetes presents chronic complications with paradoxical symptoms, such as neuropathic pain and sensory loss. Clinical studies have revealed that thermal hyperalgesia is observed in patients with mild DPN, whereas sensory loss or hypoalgesia occurs in patients with advanced-stage diabetes [42,43]. The clinical pathogenesis underlying diabetic neuropathic pain remains elusive, although several mechanisms have been proposed, including the involvement of hemodynamic factors and thalamic neuronal dysfunction [44,45,46]. On the other hand, multiple mechanisms have been proposed in studies using diabetic rodent models, including aldose reductase/AR [47,48], high mobility group box 1/HMGB1- receptor for advanced glycation end-product/RAGE signaling [49,50], protein kinase C/PKC [51,52], poly(ADP-ribose) polymerase/PARP [53,54] and oxidative stress [55,56,57]. These are all presumed to play critical roles in the activation through primary sensory neurons and the DRG.

Several studies have used rodent models to evaluate hypoalgesia. Thermal hypoalgesia in rodents has been observed in the late stage of STZ-induced type I diabetic models [47,51,58,59,60,61,62] and in various type II diabetic models, including the HFD-induced diabetic model and db/db mice [63,64,65]. Here, we confirmed the paradoxical alteration of pain perception, such as hyperalgesia and hypoalgesia/hypoesthesia in the type II HFD- or db/db model and type I STZ-model. Notably, many studies revealed that hyperalgesia occurs at the early stage after STZ-treatment(s), and hypoalgesia does several weeks later, though the timing showing the transition from hyperalgesia to hypoalgesia depends on the experimental models [47,51,58,61,62]. In the present study, we observed significant thermal hyperalgesia in HFD-, db/db- and STZ-models, but failed to detect the thermal hypoalgesia by the time of 10 weeks after the start of HFD-treatments, or 49 days (7 weeks) after STZ-treatment. However, when we used an EPW test using 5 Hz electrical stimulation to the paw to stimulate C-fibers [14,38], significant hypoalgesia was observed at 6 weeks after the HFD, at 8 weeks-old db/db/mice or 21 days (3 weeks) after the STZ. As the EPW test using 2000 and 250 Hz electrical stimulation to stimulate Aβ and Aδ showed significant hyperalgesia throughout experiments, fiber-specific mechanisms underlying hyperalgesia and hypoalgesia seem to be more important than the time-dependent functional shift of sensory fibers. This view may be supported by the report that the spinal dorsal horn expresses high levels of TRPV1 as a molecular marker of C-fibers at the early hyperalgesia stage in STZ-treated mice, while low levels at the later hypoalgesia stage [60]. Furthermore, it is also supported by the report that microneurographic recording showed an abnormal function of C-fibers in diabetic patients [66].

Both hyperalgesia and hypoalgesia in all three of types I and II diabetic mouse models were completely abolished in LPA_1_^−/−^ and LPA_3_^−/−^ mice. In addition, all these abnormal pain behaviors in HFD- and db/db-mice were slowly reversed by repeated treatments with LPA_1/3_ antagonist, which has no acute anti-hyperalgesic action, as seen in the case of partial sciatic nerve ligation (pSNL)-induced neuropathic pain model [15]. In the present study, we chose the selection of doses and frequency of administration based on preliminary experiments and previous experiences using AM966 and Ki-16425. Specifically, AM966 was administered once per day to avoid multiple central administrations [18]. On the other hand, as the treatments with Ki16425 (30 mg/kg, s.c.) once per day showed weaker effects in the pSNL-induced neuropathic pain model, possibly due to poor brain penetrability and chemical stability in vivo, and Ki16425 at 100 mg/kg (s.c.) has some toxic effects, we performed the treatments at 30 mg/kg (s.c.) twice per day [15]. Although it is difficult to conclude due to the limited condition of treatments, the slow recovery may be related to the mechanisms underlying feed-forward amplification of LPA_3_-mediated LPA production and feed-forward functional activation of LPA_1_-mediated peripheral mechanisms, comprised of dorsal root demyelination and Ca_v_α2δ1 upregulation in DRG [12]. Thus, it seems to be a time-consuming process to exclusively block multiple mechanisms of LPA_1_- and LPA_3_-mediated feed-forward systems. It should be noted that the complete reversal of thermal hyperalgesia in LPA_1_^−/−^ and LPA_3_^−/−^ mice was observed during the late stage at 28 days after the STZ injection, but not during the initial 3-21 days. A similar delayed recovery was observed in LPA_1_^−/−^ and LPA_3_^−/−^ mice in the case of hypersensitivity, hyperalgesia, and hypoalgesia using 2000, 250, and 5 Hz of electrical stimulation, respectively, suggesting that LPA receptor-independent mechanisms may also contribute to or initiate the type I diabetic neuropathic pain. As the simultaneous and intense pain signals on sensory fibers are required for the initial production of LPA in the spinal cord, followed by self-amplification of LPA production [67], various painful machineries following the STZ-induced pancreatic destruction may contribute to the initial LPA production. These LPA receptor-independent mechanisms may include AR, PARP, and oxidative stress [47,48,53,57], though no experimental evidence is available. On the other hand, hypoalgesia was observed at the late stage, but it was completely abolished in LPA_1_^−/−^ and LPA_3_^−/−^ mice in the STZ model, as well as hyperalgesia, mechanisms underlying hypoalgesia seems to be common to the case with hyperalgesia. One of the plausible mechanisms would be the LPA-induced retraction of C-fibers, followed by the intrusion of sprouts from demyelinated A-fibers and abnormal synaptogenesis to the target spinal neurons, which was originally innervated by C-fibers, as previously proposed [5]. Thus, the unique C-fiber-related hypoalgesia as well as known hyperalgesia through LPA_1_ and LPA_3_-mediated mechanisms seems to be common in type I and type II diabetic models, which are caused by the destruction of pancreatic insulin-containing β-cells and closely related to metabolic disease, respectively. Type I diabetes mellitus is now efficiently treated by insulin injection, while type II one requires long-term treatments, and thereby comorbid abnormal pain would be more problematic. In the present study, we have obtained promising results that some LPA_1/3_ antagonists have anti-hyperalgesic actions in the type II diabetic model, though the study to evaluate the anti-hyperalgesic effects of LPA_1/3_ antagonists in the type I model remains to be performed. Based on the present findings, detailed studies using better compounds to block LPA_1_ and LPA_3_ receptor activation in the type I model as well as type II one would be expected as the next subjects.

Throughout the present study, we happened to find that the body weight gain in the HFD model was largely abolished in LPA_1_^−/−^ and LPA_3_^−/−^ mice. Some detailed analyses demonstrated that adipose growth at epididymal and perirenal regions were significantly inhibited in LPA_1_^−/−^, but not LPA_3_^−/−^ mice. On the other hand, HFD-induced increase in body and liver weights were attenuated in LPA_1_^−/−^ and LPA_3_^−/−^ mice, compared with the weights in WT mice. Although the mechanisms underlying the attenuated body, organ, and tissue weight gains in the LPA^−/−^ mice remain elusive, a report has suggested that LPA_1_ is involved in adipose growth [68]. However, the alteration of body weight or condition of obesity does not seem to be related to the LPA_1_ and LPA_3_-mediated diabetic abnormal pain behaviors, since the HFD-fed and db/db mice have increased body weight, while STZ-treated mice have decreased one. Furthermore, the LPA1/3 antagonist-treatments reversed the abnormal pain behaviors, but not the body weight gain or loss.

In conclusion, the present study demonstrated that type I and II diabetic neuropathic hyperalgesia and hypoalgesia are mediated by LPA_1_- and LPA_3_-mediated mechanisms, preceded by their receptor-independent mechanisms. Moreover, LPA receptor signaling may be a potential therapeutic target for the treatment of abnormal diabetic pain disorders.

## Figures and Tables

**Figure 1 cells-09-01906-f001:**
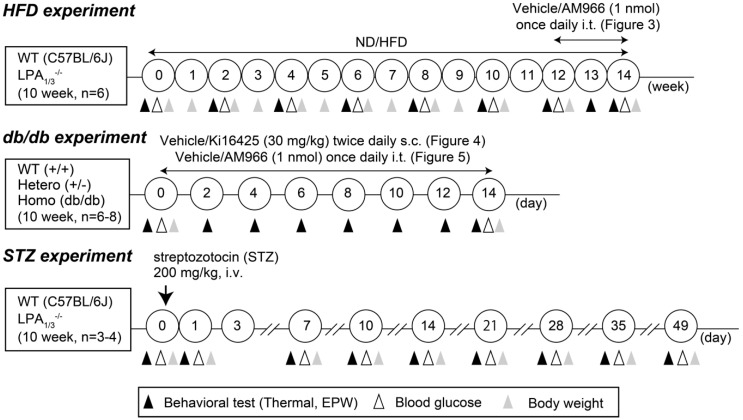
Schematized study of design.

**Figure 2 cells-09-01906-f002:**
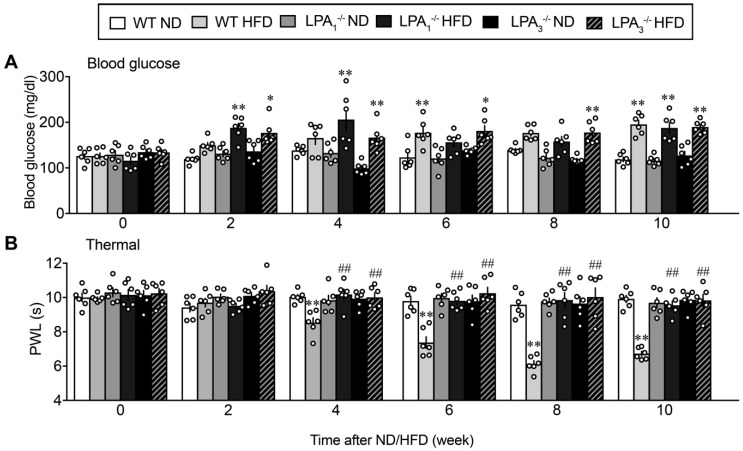
Lysophosphatidic acid (LPA) receptor 1 and 3-mediated thermal hyperalgesia in high fat diet (HFD) model sensory fiber-specific hyper- and hypoalgesia in HFD model. (**A**) Time-dependent changes in blood glucose levels of wild-type (WT), LPA_1_^−/−^ and LPA_3_^−/−^ mice after the start of normal diet (ND) or HFD feeding. (**B**) Time course of thermal hyperalgesia in WT, LPA_1_^−/−^ and LPA_3_^−/−^ mice after the start of ND or HFD feeding. * *p* < 0.05, ** *p* < 0.01, vs. normal diet (ND) group at each kind of mouse. ^##^
*p* < 0.01, vs. WT HFD, two-way ANOVA followed by Tukey’s multiple comparisons test (*n* = 6).

**Figure 3 cells-09-01906-f003:**
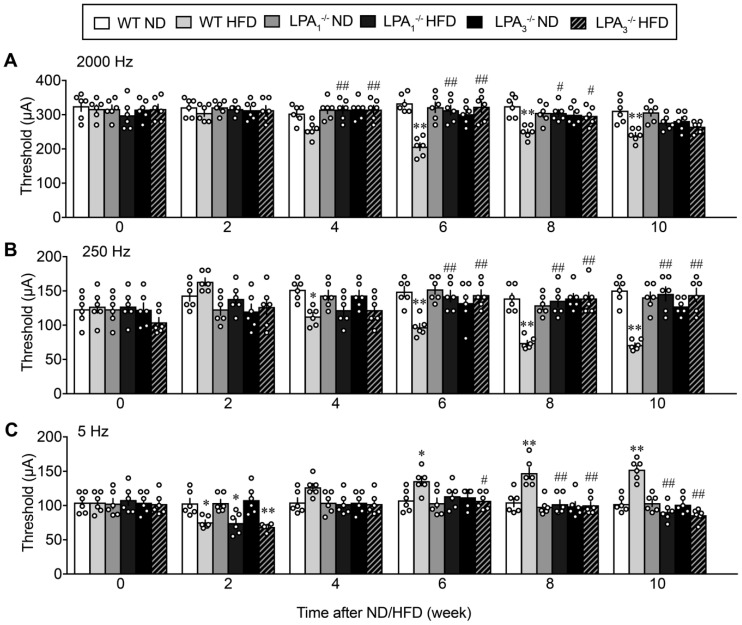
LPA receptor 1 and 3-mediated sensory fiber-specific abnormal pain behaviors in HFD model. (**A**–**C**) Time course of abnormal pain behaviors with EPW test in WT, LPA_1_^−/−^ and LPA_3_^−/−^ mice after the start of ND or HFD feeding. Frequencies of electrical stimulation were 2000 (**A**), 250 (**B**) and 5 Hz (**C**). ** *p* < 0.01, * *p* < 0.05, vs. WT ND, ^##^
*p* < 0.01, ^#^
*p* < 0.05, vs. WT HFD, two-way ANOVA followed by Tukey’s multiple comparisons test (*n* = 6).

**Figure 4 cells-09-01906-f004:**
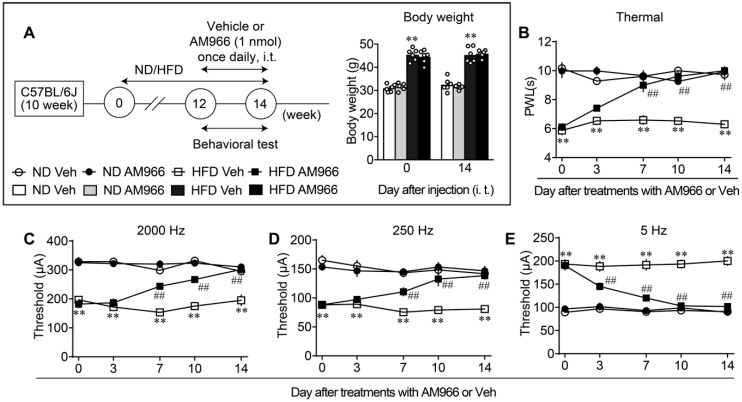
Complete reversal of abnormal pain behaviors in HFD model by intrathecal LPA_1/3_ antagonist. (**A**) Experimental schedule and lack of effects on the body weight by repeated treatments with AM966. (**B**–**E**) Time dependent reversal of nociceptive thresholds by AM966 treatments. Results represent the threshold of paw withdrawal latency (PWL; s) in the thermal test (**B**), and currents (μA) in the Electrical Stimulation-Induced Paw Withdrawal (EPW) test using 2000 (**C**), 250 (**D**) and 5 Hz (**E**). ** *p* < 0.01, vs. normal diet (ND) vehicle (Veh), ^##^
*p* < 0.01, vs. high fat diet (HFD) Veh, three-way ANOVA followed by Tukey’s multiple comparisons test (*n* = 6).

**Figure 5 cells-09-01906-f005:**
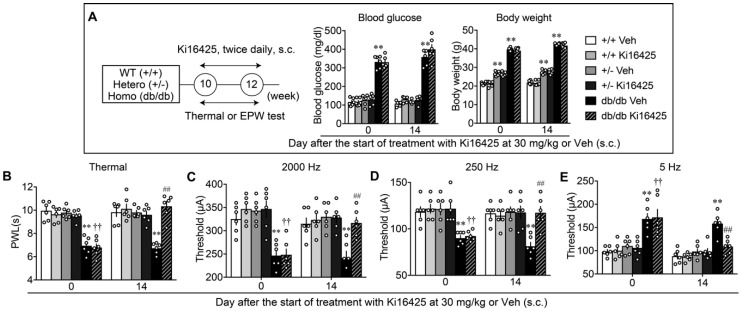
Reversal of abnormal pain behaviors in db/db mice by systemic treatments with LPA_1/3_ antagonist. (**A**) Experimental schedule and lack of effects on the blood glucose levels and body weight by repeated treatments with Ki16425 (s.c.). (**B**–**E**) Effects of repeated treatments with Ki16425 on abnormal pain behaviors in WT, leptin-deficient heterozygotes (+/–) and db/db mice. Results represent the threshold of PWL (s) in the thermal test (**B**), and currents (μA) in the EPW test using 2000 (**C**), 250 (**D**) and 5 Hz (**E**). ** *p* < 0.01, vs. +/+ Veh, ^††^
*p* < 0.01, vs. +/+ Ki16425, ^##^
*p* < 0.01, vs. db/db Veh, two-way ANOVA followed by Tukey’s multiple comparisons test (*n* = 6).

**Figure 6 cells-09-01906-f006:**
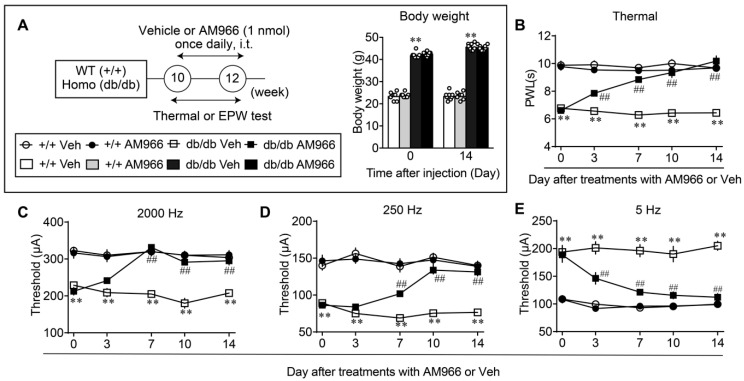
Reversal of abnormal pain behaviors in db/db mice by intrathecal treatments with LPA_1/3_ antagonist. (**A**) Experimental schedule and lack of effects on the body weight by repeated treatments with AM966 (i.t.). (**B**–**E**) Effects of repeated treatments with AM966 on abnormal pain behaviors in WT and db/db mice. ** *p* < 0.01, vs. +/+ Veh, ^##^
*p* < 0.01, vs. db/db Veh, three-way ANOVA followed by Tukey’s multiple comparisons test (*n* = 8).

**Figure 7 cells-09-01906-f007:**
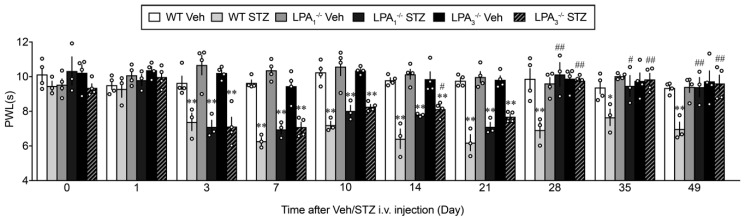
LPA receptor 1 and 3-mediated thermal hyperalgesia in type I STZ-induced diabetic mouse model. Time-dependent changes in the latency to show paw withdrawal behaviors in WT, LPA_1_^−/−^ and LPA_3_^−/−^ mice after Veh or STZ injection. * *p* < 0.05, ** *p* < 0.01, vs. Veh group at each kind of mouse, ^##^
*p* < 0.01, ^#^
*p* < 0.05, vs. WT STZ, two-way ANOVA followed by Tukey’s multiple comparisons test (WT Veh, LPA_1_^−/−^ Veh, LPA_3_^−/−^ Veh and STZ, *n* = 4; WT STZ, LPA_1_^−/−^ STZ, *n* = 3).

**Figure 8 cells-09-01906-f008:**
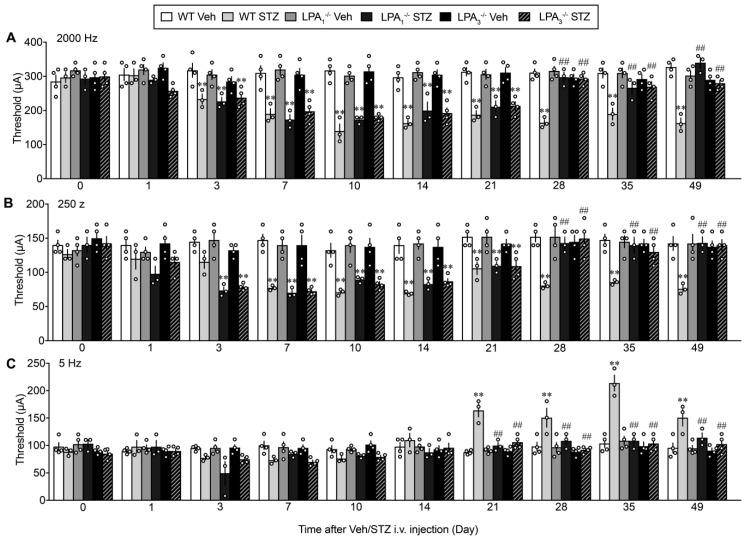
LPA receptor 1 and 3-mediated sensory fiber-specific hyper- and hypoalgesia in STZ model. (**A**–**C**) Time-dependent changes in the threshold of electrical stimulation to cause nociceptive withdrawal behaviors in WT, LPA_1_^−/−^ and LPA_3_^−/−^ mice after the Veh or STZ injection. Electrical stimulation was given at 2000 (**A**), 250 (**B**) and 5 Hz (**C**). ** *p* < 0.01, vs. Veh at each kind of mice, ^##^
*p* < 0.01, vs. WT STZ, two-way ANOVA followed by Tukey’s multiple comparisons test (WT Veh, LPA_1_^−/−^ Veh, LPA_3_^−/−^ Veh and STZ, *n* = 4; WT STZ, LPA_1_^−/−^ STZ, *n* = 3).

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
