# Peer review of "Lysophosphatidic Acid Receptor 1- and 3-Mediated Hyperalgesia and Hypoalgesia in Diabetic Neuropathic Pain Models in Mice"

_cells, 2020, doi:10.3390/cells9081906_

Round 1

Reviewer 1 Report

In this paper the authors describe the involvement of LPA1 and LPA3 receptors in diabetic neuropathic pain mouse models. Diabetic peripheral neuropathy (DPN) is one of the most frequent complications of type I (T1D) and type II diabetes (T2D). However, the current pharmacological treatment for this condition has several associated undesirable side effects that limit the patient long-term compliance. Hence, novel therapeutic options are highly desirable in this area. In this context, it has been described that LPA1 and LPA3 play important roles in the initiation and maintenance of neuropathic pain, so the authors’ aim is to establish the involvement of these two receptors in DPN so that they can be considered therapeutic targets of interest for treating this condition.

To establish the role of LPA1 an LPA3 receptors in DPN the authors develop T1D and T2D models in LPA1-/- and LPA3-/- mice and compare their phenotypes with the same models in wild type mice. Once the appropriate model has been established and characterized, they also study the effect of two antagonists, AM966 (selective LPA1 antagonist) and Ki16425 (non-selective LPA1-3 antagonist) in T2D models. Their results suggest that LPA1 and LPA3 receptors are indeed involved in the abnormal pain behaviors observed in T1D and T2D patients and accordingly they could be interesting targets for the development of new drugs for treating DPN. Overall, the research topic (DPN and lysophosphatidic acid signaling) is important and of current interest. The manuscript is clearly written, the methodology is technically sound and the conclusions are supported by the data. However, some aspects need to be addressed before the work can be published in Cells journal.

Major modifications required:

-The authors should justify the selection of the different diabetes models used.

-It is not clear why the authors do not use Ki16425 in the HFD model and only assay the compound AM966. If the interest is to establish the role of LPA3, Ki16425 should also be used.

-The authors should report the effects of Ki16425 and AM966 in the T1D STZ model.

-The dose in mg/kg of both compounds should be given to facilitate comparison. In addition, the selectivity profile of both compounds should be included in the manuscript.

-The selection of the dose must be explained and also how the dose used is translated into the in vivo concentration and its corresponding receptor IC50 or Ki values.

-The frequency of administration (twice daily, once daily) and the route of administration should be also discussed.

-Limitations of the used antagonists (that could affect the conclusions of the work) should be discussed in the manuscript (ie, low stability, low solubility, low receptor subtype selectivity…)

-Recent work of other groups regarding the involvement of LPA signaling and, in particular, of the role of LPA1 in NP should be mentioned in the references (for example, J. Med. Chem. 2020, 63, 2372).

Author Response

Please see the attactment.

Reviewer 2 Report

The authors investigated the role of Lysophosphatidic acid (LPA) signaling, in particular the involvement of LPA1 and LPA3 signaling, in the development of neuropathic pain in type I and II diabetes models. They found that LPA1 and LPA3 signaling mediate type I and II diabetic neuropathic hyperalgesia and hypoalgesia. In fact, in all the experimental models studied, the LPA1 - / - and LPA3 - / - mice do not present either hyperalgesia or hypoalgesia. Therefore, LPA1 and LPA3 may represent efficacious therapeutic targets for the reduction of abnormal diabetic neuropathic pain. The study is interesting, well organized and articulated in its experimental evaluations. However

1) In Section 2, the Materials and Methods are well described but it would be advisable to indicate the number of animals present in each experimental group and schematize the study design, reporting the various treatments and the duration of each, in such a way as to have a summary picture easily and clearly accessible.

2) Have the authors evaluated other metabolic parameters, such as triglycerides and cholesterol since their increase in HFD diet is known? Could the presence of lipid drops in the cytoplasm influence the test response?

3) In Section 3.4 lines 230-234 Figure 4A should be explained more clearly regarding treatment.

Minor review:

In the supplementary figures file in the legend of Figure S4: A-C refers to body weight and not to blood glucose levels; D-F refers to blood glucose levels and not body weight.

In the text:

page 8 line 267: Figure S4A is not the blood glucose level but is Figure S4D;

page 8 line 270: Figure S4B,C is not the blood glucose level but is Figure S4E,F;

page 8 line 272: Figure S4D is not body weight but is Figure S4A;

page 8 line 273: Figure S4E,F is not body weight but is Figure S4B,C.

Round 2

Reviewer 1 Report

The new revised version can be published in its current form.